# Estimating Permutation Entropy Variability via Surrogate Time Series

**DOI:** 10.3390/e24070853

**Published:** 2022-06-22

**Authors:** Leonardo Ricci, Alessio Perinelli

**Affiliations:** 1Department of Physics, University of Trento, 38123 Trento, Italy; 2CIMeC, Center for Mind/Brain Sciences, University of Trento, 38068 Rovereto, Italy

**Keywords:** permutation entropy, uncertainty estimation, surrogate generation

## Abstract

In the last decade permutation entropy (PE) has become a popular tool to analyze the degree of randomness within a time series. In typical applications, changes in the dynamics of a source are inferred by observing changes of PE computed on different time series generated by that source. However, most works neglect the crucial question related to the statistical significance of these changes. The main reason probably lies in the difficulty of assessing, out of a single time series, not only the PE value, but also its uncertainty. In this paper we propose a method to overcome this issue by using generation of surrogate time series. The analysis conducted on both synthetic and experimental time series shows the reliability of the approach, which can be promptly implemented by means of widely available numerical tools. The method is computationally affordable for a broad range of users.

## 1. Introduction

Information measures are increasingly important in the investigation of complex dynamics that underlie processes that occur in different frameworks [1]. The task mostly consists of evaluating the amount of information out of a discrete sequence of a given length. The sequence can be made of symbols belonging to a finite alphabet, e.g., DNA sequences [2], or correspond to a time series of realizations of a physical random variable, which ideally takes on values out of a continuous set. The latter case can be, at the expense of a loss in resolution, reduced to the former one by encoding trajectories into symbols upon a suitable coarse graining of the state space [3,4]. The information content of a sequence built on a finite alphabet can be then inferred by applying techniques such as Shannon block entropy [2,4,5,6].

In time series analysis, permutation entropy (PE), first devised in 2002 by Bandt and Pompe [7], provides an alternative approach. Possibly due to the symbolic sequences being *naturally* generated by the variable of interest without *further model assumptions* [7] or any *preassigned partition of the phase space* [8], PE has progressively become one of the most used information measures. Given a time series and a positive, integer dimension *m*, PE is estimated by first encoding *m*-dimensional, consecutive sections of that time series into symbolic sequences corresponding to permutations, and then by counting their occurrences. PE finally corresponds to the plug-in Shannon entropy computed on the sample distribution of the observed occurrence frequencies. Due to the simplicity in its evaluation, PE has become popular in many research areas ranging from medicine [9,10,11] to neuroscience [12,13,14,15], from climatology [16] to optoelectronics [17,18] and even transport complexity analysis [19].

A major property of PE is the fact that its growth rate with the dimension *m* asymptotically coincides with the Kolmogorov–Sinai entropy of the time series source [7]: the larger the *m*, the more reliable the measure. Unfortunately, given *m*, the number of the possible symbolic sequences can be as large as m!. For this reason, the required computational load typically becomes cumbersome when *m* exceeds 10. In addition, in order to yield reliable sample distributions and avoid finite-size bias [20], the length of the input time series has to be much larger than the size of the set of the visited symbolic sequences.

The difficulty of reliably assessing PE at large values of the dimension *m* presents a likely reason why PE, rather than being employed as an estimator of entropy rate, is often used at fixed *m* as an irregularity or complexity marker. Indeed, PE reaches the maximum value log(m!) (throughout the paper we use “nat” units) in the case of purely stochastic time series, and the minimum value, zero, in the case of monotonic functions. More interestingly, the variation—again at fixed *m*—of PE computed on different segments of a time series can provide a marker of different states of the underlying system, i.e., of a nonstationary behavior. Cao et al. [21] first described the use of PE to detect dynamical changes in complex time series. In a sample application, they showed how the method could be used to detect the onset of epileptic seizures out of clinical electroencephalographic (EEG) recordings. The approach has been since successfully applied in a wide spectrum of fields, ranging from economics [22,23] to neurophysiology [24,25,26,27] and geophysics [28,29].

Due to it being a statistic computed on sample time series, detecting statistically significant differences of PE in different segments immediately calls for the necessity of estimating the variability, or uncertainty, of the PE computed on each single segment. (We assume here that, as far as PE is concerned, each segment is stationary. Needless to say, characterizing a segment as being stationary requires the same statistical tools to detect changes. In other words, addressing the stationary requirement can promptly lead to circular reasoning.) A standard approach consists of observing different time series produced under the very same conditions and then applying basic statistics to assess the uncertainty of PE. Unfortunately, in fields such as geophysics and astrophysics, experimental conditions are neither controllable nor reproducible. Reproducibility is also a major issue in life sciences; a way to circumvent it consists of observing samples recorded from different individuals, under the rather strong assumption that they provide similar responses [27,30,31].

An alternative approach to the assessment of the uncertainty of sample PE relies on the symbolic sequence generation being approximated as a Markov process [32,33]. The inference of the corresponding stochastic matrix would analytically lead to the evaluation of the uncertainty out of a single measurement [34], a special case thereof occurring when the process is memoryless [35]. Alternatively, the uncertainty can be estimated by bootstrap methods [36]. Unfortunately, the number of nonzero elements of a stochastic matrix describing the generation of symbolic sequences with dimension *m* can be as large as m·m! (each one of the m! symbolic sequences can be followed by up to *m* symbolic sequences), thus making a reliable inference of each element of the matrix mostly unpractical.

In this paper, we investigate an alternative method to phenomenologically assess the PE variability out of the single time series which the PE was computed on. The method exploits the generation of surrogate time series, a randomization technique that is commonly used to conduct data-driven hypothesis testing [37,38]. We rely here on the iterative amplitude-adjusted Fourier transform (IAAFT) algorithm [39], possibly the most reliable among the methods that address continuous processes [38]. Both PE and IAAFT can be promptly implemented by using numerical packages that are available from open source repositories (more details are provided in Section 2 and Section 3).

To check the robustness of the method, we consider time series generated by two test bench systems: the chaotic Lorenz attractor with different degrees of observational noise, and an autoregressive process with different autocorrelation time values. The use of both these systems provides a reliable emulation of a wide range of real experimental situations. Our analysis shows that the variability of PE can be indeed reliably estimated on a single time series. We then applied the method on an experimental case that concerns the recognition, via analysis of EEG recordings, of cognitive states corresponding to closed and opened eyes in resting state. Furthermore, in this case, the method definitely stood the test.

The work is organized as follows. A summary of PE and its variability is presented in Section 2, whereas surrogate generation is the topic of Section 3. In Section 4, we discuss the surrogate-based estimation of PE variability on time series produced by two synthetic systems. The application of the method to experimental time series is addressed in Section 5. Final remarks are discussed in Section 6.

## 2. Permutation Entropy and Its Variability

Given a scalar time series $X=\{x_{n}\}$, let the vector $x_{n}\equiv \left(x_{n},x_{n+1},\cdots ,x_{n+m-1}\right)$ be a window, or trajectory, comprised of *m* consecutive observations. Following Bandt and Pompe [7], the window $x_{n}$ is encoded as the permutation, $\left(s_{n,1},s_{n,2},\cdots ,s_{n,m}\right)$, where each number $s_{n,j}$ (1⩽j⩽m) is an integer that belongs to the range [1,m] and corresponds to the rank, from the smallest to the largest, of $x_{n+j-1}$ within $x_{n}$. Ties between two or more *x* values are solved by assigning the relative rank according to the index *j*. Let ${\hat{p}}_{S}$ be the rate with which a symbolic sequence *S* is observed within a sufficiently long time series *X*. The sample PE of *X*, computed by setting the symbolic sequence dimension *m*, is defined as
(1)$${\hat{H}}_{m}\left(X\right)=-\sum_{\left\{S\right\}}\left({\hat{p}}_{S}\log{\hat{p}}_{S}\right)+\frac{\hat{M}-1}{2(N-m+1)}\;,$$
where the sum, corresponding to the so-called plug-in estimator, runs over the set {S} of the visited permutations of *m* distinct numbers, whose size $\hat{M}$ satisfies $\hat{M}⩽m!$. In Equation (Equation 1), the additional term $\frac{\hat{M}-1}{2(N-m+1)}$ is instead the so-called Miller–Madow correction [40,41,42,43], which compensates for the negative bias affecting the plug-in estimator. Software packages that implement PE are available from open source repositories [44,45,46].

In Equation (Equation 1) the symbol $\hat{\cdot }$ on both $p_{S}$ and $H_{m}$ expresses their being sample statistics, which makes the sample PE ${\hat{H}}_{m}\left(X\right)$ a quantity affected by an uncertainty $\sigma_{H_{m}\left(X\right)}$. The uncertainty $\sigma_{H_{m}\left(X\right)}$ is crucial in order to express the significance of the sample value given by Equation (Equation 1). As discussed in Section 1, a special case in which the uncertainty $\sigma_{H_{m}\left(X\right)}$ can be estimated out of the set $\left\{{\hat{p}}_{S}\right\}$ occurs when the symbolic sequence generation is memoryless [35]. However, besides the fact that most dynamical systems do have a memory, the standard PE encoding procedure requires overlapping trajectories, which intrinsically produces a memory effect on the succession of symbols [32]. Thus, for example in the case m=3, the permutation (1,2,3) can be only followed by itself or by (1,3,2), (3,1,2).

Nevertheless, for the sake of comparison, it is worth evaluating the uncertainty $\sigma_{H_{m},0}$ that would occur in the memoryless case. As shown by Basharin [40], the variance of sample Shannon entropy ${\hat{H}}_{\mathrm{Shannon}}$ computed on an *N*-fold realization of a memoryless, multinomial process described by the set of probabilities $\left\{p_{i}\right\}$, scales with *N* as
$$\sigma_{{\hat{H}}_{\mathrm{Shannon}}}^{2}=\frac{\Lambda_{0}}{N}+O\left(\frac{1}{N^{2}}\right)\;,$$
where the parameter $\Lambda_{0}$, defined as $\Lambda_{0}=\sum_{i}\left(p_{i}\log^{2}p_{i}\right)-{\left(\sum_{i}p_{i}\log p_{i}\right)}^{2}$, is a sort of population variance of the random variable $-\log p_{i}$. Applying this scaling behavior to PE and defining the plug-in estimator ${\hat{\Lambda }}_{0}$ of the parameter $\Lambda_{0}$ as [34,35]
$${\hat{\Lambda }}_{0}\left(X\right)=\sum_{\left\{S\right\}}\left({\hat{p}}_{S}\log^{2}{\hat{p}}_{S}\right)-{\left(\sum_{\left\{S\right\}}{\hat{p}}_{S}\log{\hat{p}}_{S}\right)}^{2}\;,$$
the uncertainty $\sigma_{H_{m},0}$ of the sample PE in the memoryless case can be estimated as
(2)$$\sigma_{H_{m},0}\left(X\right)\approx \sqrt{\frac{{\hat{\Lambda }}_{0}\left(X\right)}{N-m+1}}\;,$$
where *N* is the length of the time series *X*.

## 3. Summary of Surrogate Generation

Given a time series, henceforth referred to as the “original” one, the goal of surrogate generation is the synthesis of a set of close replicas that share basic statistical properties with the original one [38]. The approach was first proposed [37] to generate data sets consistent with the null hypothesis of an underlying linear process, with the ultimate aim of testing the presence of nonlinearity. The method was then generalized to test other null hypotheses, most notably to evaluate the statistical significance of cross-correlation estimates [47,48,49], as well as other coupling metrics such as transfer entropy [50].

The implementation of surrogate generation requires the algorithm to mirror the null hypothesis to be tested. Therefore, a straightforward random shuffling of data points preserves the amplitude distribution of the original time series while destroying any temporal structure, and thus allows for testing the null hypothesis of a white noise source having a given amplitude distribution [37]. Another implementation targets the null hypothesis of an underlying Gaussian process with a given finite autocorrelation. In this case, the surrogate generation is conducted by Fourier-transforming the original time series and then randomizing the phase of the resulting frequency domain sequence: by virtue of the Wiener–Kinchine theorem, computing the inverse Fourier transform of this last sequence leads to a time series that has the same autocorrelation as the original one [37]. While the exact, simultaneous conservation of both the amplitude distribution and the autocorrelation function is impossible, the IAAFT algorithm conserves the amplitude distribution while approximating the spectrum and thus the autocorrelation [39].

A more general way of producing surrogate data consists of setting up a “simulated annealing” pipeline in which, at each step, time series samples are randomly swapped and, depending on the effect of the swap on a suitably defined cost function, the step is either accepted or rejected [51]. This way, the statistical properties to be preserved in surrogate data—and thus the details of the null hypothesis to be tested—are transferred from the algorithm to the definition of the cost function, so that in principle *any* hypothesis can be tested. However, this versatility comes at the expense of increased computational costs [38].

In the present work, the IAAFT algorithm for surrogate generation, first described by Schreiber and Schmitz [39] and possibly the most reliable one in the case of continuous processes, was used. The main feature of IAAFT is the simultaneous quasi-conservation of the amplitude distribution and the autocorrelation. Consequently, the local structure of the trajectories, and thus the statistical properties of the encoded symbolic sequences, is expected to be preserved. On the contrary, possibly the simplest surrogate generation method, namely a random shuffling of data points, would directly act on the symbolic sequences similarly to the superposition of white noise. It has to be stressed that, while the analysis described below proves IAAFT to be satisfactory, the choice of an optimal surrogate generation method can be a matter of further investigation, as discussed in Section 6.

The main steps of the algorithm are summarized below [52]. Software packages that implement IAAFT are available from open source repositories [53,54,55,56,57]. An implementation in Matlab and Python, developed by the authors, is also available [58].

Given the *n*-th value $x_{n}$ of a time series $X=\{x_{n}\}$, let $r_{n}$ be the amplitude rank, from the smallest to the largest, of $x_{n}$ within the time series *X*.

In step 0 of the algorithm, the values of the time series *X* are randomly shuffled so as to yield a sequence $Y_{0}=\left\{y_{0,n}\right\}$.Step 1 consists of FFT-transforming the two time series *X*, $Y_{0}$, thus producing the frequency domain sequences $\left\{{\tilde{X}}_{k}\right\}$, $\left\{{\tilde{Y}}_{0,k}\right\}$, respectively. Let $\phi_{0,k}$ be the phase, i.e., the argument, of the complex number ${\tilde{Y}}_{0,k}$ (${\tilde{Y}}_{0,k}=\left|{\tilde{Y}}_{0,k}\right|e^{i\phi_{0,k}}$).Step 2 consists of two parts. First, the two Fourier sequences $\left\{{\tilde{X}}_{k}\right\}$, $\left\{{\tilde{Y}}_{0,k}\right\}$ are mingled together to produce the sequence $\left\{{\tilde{Z}}_{0,k}\right\}$, where ${\tilde{Z}}_{0,k}=\left|{\tilde{X}}_{k}\right|e^{i\phi_{0,k}}$; in other words, ${\tilde{Z}}_{0,k}$ has the amplitude of ${\tilde{X}}_{k}$ and the phase of ${\tilde{Y}}_{0,k}$. Second, the inverse Fourier transform of the sequence $\left\{{\tilde{Z}}_{0,k}\right\}$ is computed, thus yielding the time-domain sequence $Z_{0}=\left\{z_{0,n}\right\}$. For each value $z_{0,n}$, let $q_{0,n}$ be the rank, from the smallest to the largest, of $z_{0,n}$ within the time series $Z_{0}$.Step 3 consists of replacing the *n*-th value $z_{0,n}$ with the value $x_{m}$ such that $r_{m}=q_{0,n}$. This step leads to the amplitude-adjusted sequence $Y_{1}=\left\{y_{1,n}\right\}$.Steps (1) to (3) are finally iteratively repeated until the *i*-th cycle characterized by
$$\sum_{n}{\left(y_{i,n}-y_{i-1,n}\right)}^{2}<10^{-6}\cdot \sum_{n}y_{i,n}^{2}\;,$$
i.e., until the Euclidean distance between the sequences $Y_{i}$, $Y_{i-1}$ (considering them as vectors) becomes sufficiently small with respect to their norms.

Let *Y* be the final sequence $Y_{i}$ produced by the iteration. By construction, each sequence $Y_{i}$, and therefore also *Y*, has the same amplitude distribution as the original one *X*. As shown by Schreiber and Schmitz [39], the iteration leads to a discrepancy between the spectra of *X* and *Y* whose order can be quantified as
$$\frac{\sum_{k}{\left(|{\tilde{Y}}_{k}\left|-\right|{\tilde{X}}_{k}|\right)}^{2}}{\sum_{k}{\left|{\tilde{X}}_{k}\right|}^{2}}∼\frac{1}{N^{3/2}}\;.$$

Despite its robustness, IAAFT is sensitive to amplitude mismatches between the beginning and the end of the input original time series [38]. This issue, referred to as periodicity artifacts, can be overcome by trimming the original time series to generate a shorter segment whose end points have sufficiently close amplitudes [37,38]. While in the case of synthetic time series the operation can be implemented without losing information (for example, arbitrarily large numbers of equivalent original time series can be produced), the same is not true for experimental time series. We therefore opted for an alternative approach that consists of detrending the time series so that the end points have equal values. To this purpose, each value $x_{n}$ of an input time series *X* is modified according to the following expression:(3)$$x_{n}^{\prime }=x_{n}-\frac{x_{N}-x_{1}}{N-1}\left(n-1\right)\;,$$
where n∈[1,N]. Henceforth, for the sake of simplicity, the resulting time series $X^{\prime }$ is renamed as *X*.

Indeed the detrending operation of Equation (Equation 3) can slightly modify the number of occurrences of the symbolic sequences and consequently the sample PE value: within the same *m*-dimensional trajectory, the end points are mutually displaced by an amount of order mσ/N, where σ is the standard deviation of the time series. However, whenever m≪N, the effects on the PE assessment are negligible.

## 4. Surrogate-Based Estimation of PE Variability in the Case of Synthetic Sequences

This section describes the details of surrogate-based estimation of PE variability. To evaluate the performance of the method we consider two synthetic dynamical systems, which allow for the generation of arbitrary numbers of similar time series. The two systems are a Lorenz attractor affected by observational noise and an autoregressive fractionally-integrated moving-average (ARFIMA) process. Both are described in Section 4.3.

### 4.1. Reference Value of PE Variability

Let *X* be the realization of a time series of length *N* generated by a dynamical system. As explained in the previous section, the time series is supposed to be detrended so as to avoid periodicity artifacts. We use the following standard statistical approach, which exploits the system being synthetic, to evaluate the reference uncertainty $\sigma_{H_{m}\left(X\right)}$ on the PE assessment ${\hat{H}}_{m}\left(X\right)$. The uncertainty $\sigma_{H_{m}\left(X\right)}$ will be then used to evaluate the reliability of the surrogate-based estimation.

Given *X*, a set of L−1 additional realizations $X_{\ell }$, with 1⩽ℓ⩽L−1 is generated out of the same system by randomly changing the starting seed and applying the detrending operation described by Equation (Equation 3). Upon setting $X_{0}\equiv X$ and evaluating the sample mean $\langle {\hat{H}}_{m}\left(X\right)\rangle$ as
$$\langle {\hat{H}}_{m}\left(X\right)\rangle =\frac{1}{L}\sum_{\ell =0}^{L-1}{\hat{H}}_{m}\left(X_{\ell }\right)\;,$$
the sample variance $s_{{\hat{H}}_{m}\left(X\right)}^{2}$ is evaluated as
$$s_{{\hat{H}}_{m}\left(X\right)}^{2}=\frac{1}{L-1}\sum_{\ell =0}^{L-1}{\left[{\hat{H}}_{m}\left(X_{\ell }\right)-\langle {\hat{H}}_{m}\left(X\right)\rangle \right]}^{2}\;.$$
The resulting sample standard deviation $s_{{\hat{H}}_{m}\left(X\right)}$ is an estimate of the uncertainty $\sigma_{H_{m}\left(X\right)}$ of the permutation entropy of the original time series *X*. In the present work, L=100.

### 4.2. Surrogate-Based Estimation of PE Variability

The uncertainty of the permutation entropy of the original sequence *X* is then estimated by relying on surrogate generation as follows. Let $\left\{Y_{\ell }\right\}$, with 1⩽ℓ⩽L (again, *L* is set to 100), a set of surrogates of the time series *X* generated via IAAFT and detrended via Equation (Equation 3).

Upon evaluating the sample mean $\langle {\hat{H}}_{m}\left(Y\right)\rangle$ as
$$\langle {\hat{H}}_{m}\left(Y\right)\rangle =\frac{1}{L}\sum_{\ell =0}^{L-1}{\hat{H}}_{m}\left(Y_{\ell }\right)\;,$$
the sample variance $s_{{\hat{H}}_{m}\left(Y\right)}^{2}$ is evaluated as
(4)$$s_{{\hat{H}}_{m}\left(Y\right)}^{2}=\frac{1}{L-1}\sum_{\ell =0}^{L-1}{\left[{\hat{H}}_{m}\left(Y_{\ell }\right)-\langle {\hat{H}}_{m}\left(Y\right)\rangle \right]}^{2}\;.$$
The resulting sample standard deviation $s_{{\hat{H}}_{m}\left(Y\right)}$ is taken as a measure of the surrogate-based variability of the permutation entropy of a time series *X*.

### 4.3. Synthetic Dynamical Systems

The noiseless Lorenz attractor is described by the following differential equation system
$$\begin{array}{ccc} \frac{dx}{dt} & = & a(y-x)\;,\\ \frac{dy}{dt} & = & x(c-z)-y\;,\\ \frac{dz}{dt} & = & xy-bz\;. \end{array}$$
Here, the three parameters are set to a=10, b=8/3, c=28. The system is integrated by means of a Runge–Kutta (8,9) Prince–Dormand algorithm, with the integration time dt=0.3 that also coincides with the sampling time. The value $x_{n}$ of the *x* coordinate computed at the *n*-th step is then added to a realization of a normal random variable with zero mean and standard deviation $\eta \cdot \sigma_{\mathrm{Lorenz}}$, where η is a non-negative amplitude and $\sigma_{\mathrm{Lorenz}}=7.9252822$ is the standard deviation of the *x* coordinate of the noiseless Lorenz attractor. One has:$$x_{n}^{\prime }=x_{n}+\sigma_{\mathrm{Lorenz}}\cdot \eta \cdot \epsilon_{n}\;,$$
where $\epsilon_{n}$ is a standard normal random variable. In the following, for the sake of simplicity, the result $x_{n}^{\prime }$ is renamed as $x_{n}$: $x_{n}^{\prime }\to x_{n}$. The case η=0 corresponds to the noiseless Lorenz attractor. For any other positive value η, the resulting signal has a power signal-to-noise ratio given by $\eta^{-2}$.

The ARFIMA process is defined as
$$x_{n}=\sum_{k=1}^{k_{\max}}\frac{\rho \;\Gamma (k-\rho )}{\Gamma (1-\rho )\;\Gamma (k+1)}\;x_{n-k}+\epsilon_{n}\;,$$
where $\epsilon_{n}$ is a standard normal random variable, $k_{\max}=100$ (ideally, $k_{\max}=\infty$), and ρ is a parameter that tunes the autocorrelation time of the process: from purely white noise in the case ρ=0, to a progressively more correlated process when ρ→1. (Negative values of ρ, which produce anti-correlated processes, are not considered here).

For both dynamical systems, integration starts from randomly chosen points. To avoid transient effects, the first $10^{3}$ steps of each integration are discarded. The resulting original time series are made of N= 10,006 points.

The chosen length is representative of many experimental situations. For example, the EEG time series analyzed in Section 5 are made of $1.5\cdot 10^{4}$ points. Indeed, the length of the time series is not expected to play a role in the present method: if the surrogate generation reliably replicates the statistical properties of the original time series, both the intrinsic variability of the original time series and those inferred by surrogate generation are expected to scale as $N^{-\frac{1}{2}}$, so that their ratio, modulo higher-order corrections, is expected to be *N*-independent.

### 4.4. Numerical Results

Before discussing the results concerning PE uncertainties, it is worth considering the PE sample means $\langle {\hat{H}}_{m}\left(X\right)\rangle$ and $\langle {\hat{H}}_{m}\left(Y\right)\rangle$ as a function of the parameters η and 1−ρ and for m=4,5,6,7. We remind that the noise content of a synthetic time series is tuned by means of the parameter η for the Lorenz system, and ρ for the ARFIMA process.

Figure 1 shows, for each system and upon a normalization by log(m!), both sample means. As expected, the normalized PE sample means approach unity as the time series become progressively more white-noise-like, i.e., when η≳0.5 (Lorenz system) and 1−ρ≳0.9 (ARFIMA process). It is worth noting that, for m=7 and for both systems, Equation (Equation 1) performs well even when the number of visited symbolic sequences approaches 7!=5040 and thus becomes comparable with the time series length of N−m+1=10000.

In the case of the ARFIMA process, the two sample means continue coinciding also at progressively longer autocorrelation times (1−ρ→0). Conversely, in the case of the Lorenz system and for $\eta ≲10^{-2}$, the values of PE evaluated on surrogate time series (yellow squares) start to significantly diverge from the values (blue dots) computed out of the set $\left\{X_{\ell }\;|\;0⩽\ell ⩽L-1\right\}$, consisting of the original synthetic time series *X* and its equivalent replicas. Indeed, surrogate data yield higher PE values than original time series, thus underpinning the deterministic nature of the system’s dynamics. The fact that no divergence is observed in the case of the ARFIMA process hints at an intrinsically noisy content of the time series also at long autocorrelation times.

In the case of the Lorenz system, the sample standard deviations $s_{{\hat{H}}_{m}\left(X\right)}$ and $s_{{\hat{H}}_{m}\left(Y\right)}$ are shown in Figure 2 as a function of the parameter η. Similarly, in the case of the ARFIMA process, $s_{{\hat{H}}_{m}\left(X\right)}$ and $s_{{\hat{H}}_{m}\left(Y\right)}$ are shown in Figure 3 as a function of the parameter 1−ρ. In both cases, results are presented for m=4,5,6,7. In addition, each figure shows the uncertainty estimate $\sigma_{H_{m},0}$ given by Equation (Equation 2) and corresponding to the memoryless (multinomial) case, as well as the Harris limit [42]. This limit is given by $\sqrt{\frac{m!-1}{2N^{2}}}$ and corresponds to the special case in which the multinomial distribution is uniform. This situation is reached in the case of sufficiently high noise, so that all symbolic sequences become equiprobable. Finally, Figure 2 and Figure 3 also show the ratio $s_{{\hat{H}}_{m}\left(Y\right)}/s_{{\hat{H}}_{m}\left(X\right)}$ between the surrogate estimates of the PE uncertainty and the related reference values.

Unless the autocorrelation time becomes too large, the ratio $s_{{\hat{H}}_{m}\left(Y\right)}/s_{{\hat{H}}_{m}\left(X\right)}$ lays within the interval [0.5,1]. This ratio being of order 1 is a remarkable result. The light underestimation of the real uncertainty of the PE assessment is likely due to the IAAFT surrogate generation producing high-fidelity replicas of the original signal, though without predicting its exact variability due to the intrinsic lack of knowledge of the signal source. Nevertheless, the level of prediction of the real PE assessment uncertainty is surprisingly good: a factor 0.5 is small enough so as to allow for reasonable estimates of the level of significance of a PE assessment.

### 4.5. Assignment of the Surrogate-Based PE Uncertainty

Considering the results of the analysis described above, the procedure to evaluate the uncertainty of a single PE assessment via IAAFT surrogate generation can be summarized as follows.

Given an input time series *X*:Compute a detrended version of *X* by applying Equation (Equation 3) of Section 3;Generate a set $\left\{Y_{\ell }\right\}$ of *L* surrogate time series via IAAFT;Upon evaluating the PE on each $Y_{n}$, compute the standard deviation $s_{{\hat{H}}_{m}\left(Y\right)}$ via Equation (Equation 4);Finally, assign the uncertainty $\Sigma_{{\hat{H}}_{m}}$ of the PE assessment on *X* by setting
$$\Sigma_{{\hat{H}}_{m}\left(X\right)}=\alpha \cdot s_{{\hat{H}}_{m}\left(Y\right)}\;.$$

We suggest L=100 and α=2. The former value is justified by ${\hat{s}}_{{\hat{H}}_{m}\left(Y\right)}$ being a sample standard deviation, and thus being affected by a relative uncertainty that approximately scales as ${\left(2L\right)}^{-1/2}$. The value L=100 is then a reasonable trade-off between the urge of obtaining a reliable estimate of ${\hat{s}}_{{\hat{H}}_{m}\left(Y\right)}$ and affordable computational costs. The latter value α=2 corresponds to the reciprocal of the ratio $s_{{\hat{H}}_{m}\left(Y\right)}/s_{{\hat{H}}_{m}\left(X\right)}$, as described at the end of Section 4.4. The setting of the parameter α is further discussed in Section 6.

## 5. Estimation of PE Variability on Experimental Time Series

In the present section, we discuss the implementation of the surrogate-based estimation of PE variability in the case of a PE analysis of experimental electrophysiological data.

The data correspond to resting-state EEG time series recorded on 30 young healthy subjects and made available on the LEMON database [59,60]. Data recording was conducted in accordance with the Declaration of Helsinki. The related study protocol was approved by the ethics committee at the medical faculty of the University of Leipzig, Germany (reference number 154/13-ff). Details of the set of subjects, the recording procedure, the preprocessing steps, and source reconstruction, are extensively described in two recent papers [61,62].

The reconstructed time series considered here correspond to the two “V1” brain areas as defined in the atlas by Glasser et al. [63], which belong to the left and right visual cortex. Each EEG acquisition session was comprised of 16 successive and interleaved segments corresponding to eyes-closed (eight segments) and eyes-opened (eight segments) conditions; recordings commenced with closed eyes. To maximize stationarity, and to therefore avoid both transient effects at the beginning and fatigue effects at the end, we neglect here the first and the last pairs of conditions. Because the recorded segments had a duration between 60 s and 90 s, the corresponding raw time series were trimmed down to 60 s each by symmetrically removing leading and trailing data points. The preprocessing resulted, for each subject, in two reconstructed time series of brain activity, each composed of two interleaved sets of six 60 s long segments. An example of a preprocessed time series corresponding to a single brain area of a subject is shown in Figure 4.

In addition to the time series, Figure 4 shows the normalized PE values with m=7 computed on each segment and the related surrogate-based uncertainty estimate on each PE value.

It is graphically apparent that the error bars on the PE assessments are approximately equal to each other and also correspond to the fluctuations of the PE values. This observation can be quantified by means of an analysis based on a least-squares fit, as follows.

Our null hypothesis consists of two assumptions. The first corresponds to the very claim of this paper, namely that the surrogate-based uncertainties, i.e., the error bars, correctly estimate the real uncertainty of the related PE values. Second, we assume that the PE values drift in time, e.g., due to fatigue or habituation, according to a quadratic law $a+bk+ck^{2}$.

Given a subject, a brain area and a condition—there are 120 different combinations—a least-squares fit of the quadratic law on the set of six points is carried out (see dashed lines in Figure 4). The fit consists of finding the parameters $\hat{a}$, $\hat{b}$, $\hat{c}$ that minimize the sum of the normalized residuals
$$S\left(a,b,c\right)=\sum_{k=1}^{6}\frac{{\left[{\hat{H}}_{m}\left(X_{k}\right)-\left(a+bk+ck^{2}\right)\right]}^{2}}{\Sigma_{{\hat{H}}_{m}\left(X_{k}\right)}^{2}}\;,$$
where $X_{k}$, with k=1,⋯,6, are the time series corresponding to the six segments.

If the null hypothesis holds, the sum $S(\hat{a},\hat{b},\hat{c})$ is to be distributed as a χ-square variable $\chi_{3}^{2}$ with ν=3 degrees of freedom. Figure 5 shows, for each one of the four *m* values 4,5,6,7, the histogram of the 120 values of $S(\hat{a},\hat{b},\hat{c})$. In all four cases, the histograms are in very good agreement with the plots of the $\chi_{3}^{2}$ probability density function $f_{\chi_{3}^{2}}\left(x\right)$ given by
(5)$$f_{\chi_{3}^{2}}\left(x\right)=\sqrt{\frac{x}{2\pi }}e^{-x/2}\;.$$
The few outliers (less than 15%, identified via p<0.01) are samples in which the quadratic description does not hold. It is worth mentioning that a similar analysis was carried out also assuming a linear time dependence for the PE drift. Furthermore, in this case, the agreement was satisfactory except for about 20% of the samples for which, again, the assumed drift law was not appropriate.

We conclude this section by stating that in the example shown in Figure 4, all data points corresponding to the eyes-closed condition have a PE value that is significantly lower than any point of the eyes-opened condition. The closest pair is given by the eyes-closed segment in the time range between 4 min and 5 min (third blue point from left), and the last, eyes-opened segment in the time range between 11 min and 12 min (sixth red point from left). The PE values of the two points are 0.487 and 0.507, respectively. Their $\Sigma_{{\hat{H}}_{m}\left(X\right)}$ uncertainties are 0.006 and 0.007, respectively. By using a standard two-sided *z* statistical test and assuming as a null hypothesis the fact that the difference is due to chance, the resulting *p*-value turns out to be about 0.03.

## 6. Discussion

In the method described in Section 4.5 the only parameter that requires an “educated guess” is the factor α that magnifies the sample standard deviation computed on a set of surrogate time series so as to provide the uncertainty value of the PE assessment. Our analysis shows that α typically ranges between 1 and 2, which is in any case a very small range to cope with. Remarkably, at least in the case of deterministic chaos provided by the Lorenz attractor contaminated by observational noise, α approaches unity in the noiseless limit or when the noise is dominant, while approaching two whenever noise and signal are comparable.

An important point to highlight is the fact that the approach described in this paper can be promptly generalized to other kinds of information measures, for example approximated entropy [64,65] and sample entropy [66], and in particular to those relying, such as PE, on symbolic sequence encoding, for which a major example is provided by Shannon block entropy [2].

A topic for further investigation is the location of the surrogate generation within the pipeline that leads from a time series to the assessment of information and its uncertainty: the approach proposed in this work implements the surrogate generation on the very input of the pipeline, namely the original time series. An alternative approach could instead operate on the symbolic sequence encoded out of the original time series, under the constraint that the chosen randomization technique preserves statistical and dynamical properties of the original encoding.

It has to be stressed that the choice of a suitable surrogate generation algorithm constitutes a core ingredient of the method proposed. For the cases dealt with in the present work, IAAFT proved to be a reliable technique: as expressed in Section 3, the simultaneous quasi-conservation of the amplitude distribution and the autocorrelation is crucial to preserve the local structure of the trajectories and thus the statistical properties of the encoded symbolic sequences. Nevertheless, the investigation of an optimal surrogate generation algorithm to be employed in a specific context is expected to make up an interesting development of the method.

An example is provided by how surrogate generation can be applied to the inference of information variability in the case of point processes, for which phase randomization or IAAFT are unsuitable [38]. A possibility consists of using approaches based on dithering of the event occurrence times [47,67] or on the observation of the joint probability distribution of consecutive inter-event intervals [52,68].

In conclusion, the method proposed in this paper provides a reliable estimation of the variability affecting a PE evaluation out of a single time series. The method, which relies on the generation of surrogate time series and can be promptly implemented by means of standard analytical tools, allows one to address issues concerning stationarity as well as statistically significant changes in the dynamics of a source via PE. The analysis conducted on the noisy Lorenz attractor and an ARFIMA process demonstrated that the method performs well both in the case of time series contaminated by observational noise [8] and in the case of noise with long autocorrelation time. Possible developments of the method concern its application to other information measures.

## Figures and Tables

**Figure 1 entropy-24-00853-f001:**
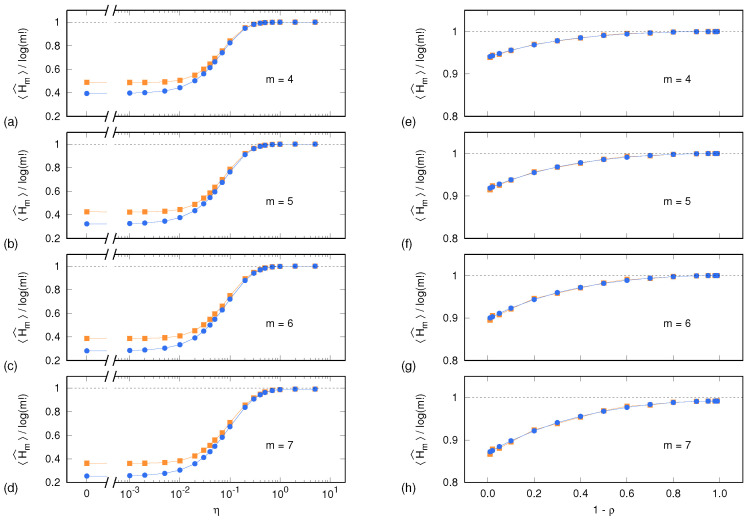
Sample mean of PE, normalized by log(m!), as a function of the tuning parameters η and 1−ρ for the Lorenz system (**a**–**d**) and for the ARFIMA process (**e**–**h**), respectively: each panel pair corresponds to a different value of the dimension *m*. Blue dots and lines correspond to the reference PE sample mean $\langle {\hat{H}}_{m}\left(X\right)\rangle$ computed out of L=100 independent realization of the synthetic time series, as described in Section 4.1. Yellow squares and lines correspond to the PE sample mean $\langle {\hat{H}}_{m}\left(Y\right)\rangle$ computed out of L=100 surrogate time series (Section 4.2). Error bars are too small to be displayed.

**Figure 2 entropy-24-00853-f002:**
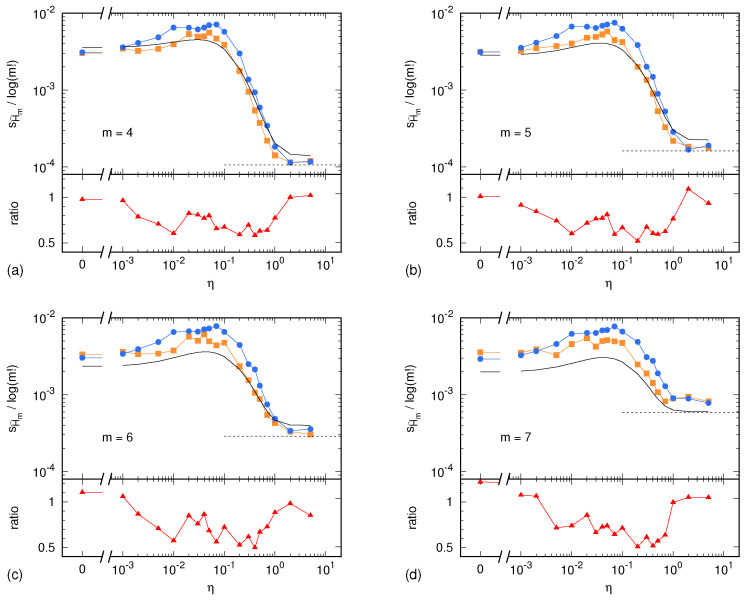
Sample standard deviation of PE, normalized by log(m!), as a function of the tuning parameter η for the Lorenz system: (**a**) m=4; (**b**) m=5; (**c**) m=6; (**d**) m=7. In the upper part of each panel, blue dots and lines correspond to the reference PE sample standard deviation $s_{{\hat{H}}_{m}\left(X\right)}$ computed out of L=100 independent realization of the synthetic time series (Section 4.1). Yellow squares and lines correspond to the PE sample standard deviation $s_{{\hat{H}}_{m}\left(Y\right)}$ computed out of L=100 surrogate time series (Section 4.2). The black, solid line corresponds to the memoryless uncertainty estimator $\sigma_{H_{m},0}$ given by Equation (Equation 2). Finally, the black, dashed line displays the Harris limit (see main text). The lower part of each panel shows the ratio $s_{{\hat{H}}_{m}\left(Y\right)}/s_{{\hat{H}}_{m}\left(X\right)}$ between the surrogate estimate of the PE uncertainty and its reference value.

**Figure 3 entropy-24-00853-f003:**
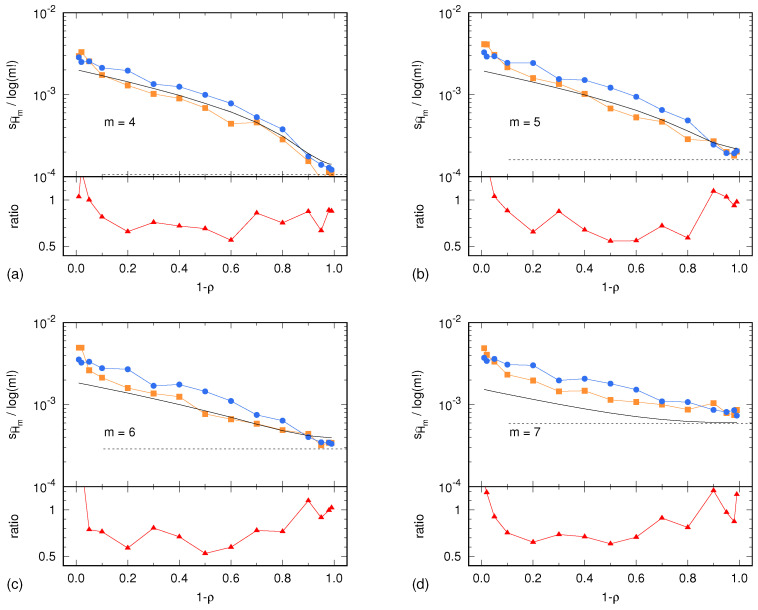
Sample standard deviation of PE, normalized by log(m!), as a function of the tuning parameter 1−ρ for the ARFIMA process. (**a**) m=4; (**b**) m=5; (**c**) m=6; (**d**) m=7. The description of the data representation (dots, lines and colors) is the same as in Figure 2.

**Figure 4 entropy-24-00853-f004:**
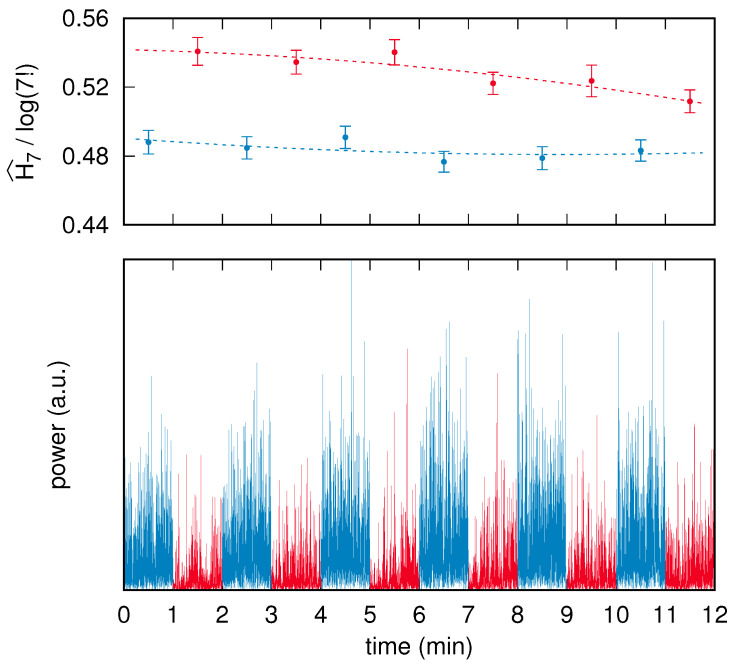
(**below**) Time series corresponding to the left visual cortex of a single subject. The blue and red segments correspond to the eyes-closed and eyes-opened conditions, respectively. (**above**) PE analysis of the segments that make up the time series: each blue (red) dot is positioned above the center of the respective eyes-closed (eyes-opened) segment plotted below, while its ordinate corresponds to the normalized PE value ${\hat{H}}_{m}\left(X\right)/\log\left(m!\right)$, with m=7, computed on that segment. Each error bar corresponds to the PE uncertainty $\Sigma_{{\hat{H}}_{m}\left(X\right)}$ assessed via the surrogate-based procedure summarized in Section 4.5. Finally, the blue (red) dashed line corresponds to the best-fit quadratic law that describes the time drift of the PE values in the eyes-closed (eyes-opened) condition.

**Figure 5 entropy-24-00853-f005:**
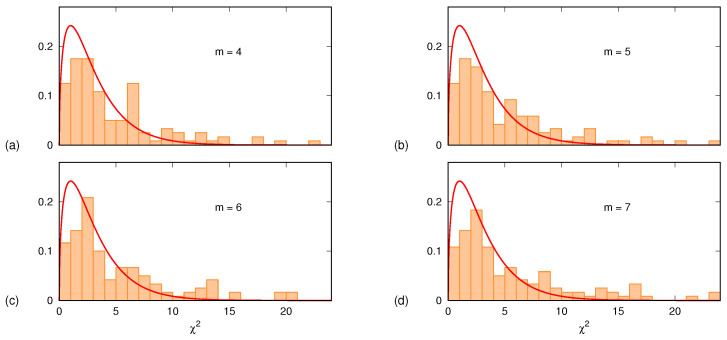
Histograms, normalized to unit area, of the sum $S(\hat{a},\hat{b},\hat{c})$, evaluated for 120 different subjects, brain areas and conditions: (**a**) m=4; (**b**) m=5; (**c**) m=6; (**d**) m=7. Each panel also displays the theoretical probability density function $f_{\chi_{3}^{2}}\left(x\right)$ of the $\chi_{3}^{2}$ random variable with ν=3 degrees of freedom, given by Equation (Equation 5).

## Data Availability

Raw EEG recordings used in the present work are available at: http://fcon_1000.projects.nitrc.org/indi/retro/MPI_LEMON.html, accessed on 23 May 2022. Preprocessed data and results are available upon request to the corresponding author.

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
