# Peer review of "Estimating Permutation Entropy Variability via Surrogate Time Series"

_entropy, 2022, doi:10.3390/e24070853_

Round 1
Reviewer 1 Report
The paper is well written and sound in its methodology and conclusions. Yet the authors, as it is custom in the literature on PE, either ignore or choose to ignore everything else except reciting the PE canon of references. I suggest it for publication with an update on this issue that will enrich the discussion and bring forth certain of the rich interrelationships among entropy notions for symbol sequences.
For example: In their second paragraph, first and second sentences, they state:
"A major property of PE is the fact that its growth rate with the dimension m asymptotically coincides with the Kolmogorov-Sinai entropy of the time series source [2]: the larger m, the more reliable the measure. Unfortunately, given m, the number of the possible symbolic sequences can be as large as m!. "
Although abstractly correct, the statement does not reflect the reality of symbol sequence analysis. PE is the Kolmogorov-Sinai entropy, asymptotically, and PE is the Shannon-Block entropy under a certain encoding.
Moreover, there is a plethora of symbol sequences (texts) that the Shannon block entropy scales as slow as "log(m)" or as a power law "m^a", (with (0<=a<1). See for example the recent reference:
“Symbolic dynamics of music from Europe and Japan” (Featured Article, SciLight) V. Basios, T. Oikonomou, R. De Gernier (2021) in "Chaos: An Interdisciplinary Journal of Nonlinear Science" 31 (5), 053122.
https://doi.org/10.1063/5.0048396
(free preprint at: https://arxiv.org/abs/1810.06119 )
And in particular the earlier references [16],[21] therein:
Basios, V. & Mac Kernan, D., [2008]. Symbolic dynamics, coarse
graining and the monitoring of complex systems, Int. J. of Bifurcation
and Chaos, 21, 3465-3475.
Ebeling, W. & Nicolis, G., [1992]. Word frequency and entropy of
symbolic sequences: A dynamical perspective, Chaos Solit. Fract. 2, 635-
650.
I strongly suggest and invite the authors to have a look and update their discussion on the obvious but unjustifiably forgotten --and so strong connection-- between PE and Shannon's Block Entropy.
Whatever one can do with PE can do, and even more, with a coarse graining and considering the Shannon Block-Entropy. This can be achieved for a great array of classes of dynamical systems and their generated texts, natural and artificial alike.
Reviewer 2 Report
Surrogate tests and permutation entropy are well-known and thoroughly investigated subjects, but this paper tackles an important issue of assessing the uncertainty of permutation entropy estimation.
The paper is excellently written. Introductory parts that present the literature survey and give a review of PE and ST are written as a tutorial and are among the best. Particularly enjoyable for reading are the authors’ comments and observations embedded into the explanations. The remainder of the paper explains the proposed method and it is also written in a straightforward and non-ambiguous way.
A coffee-break discussion at a conference would be more beneficial for both sides. There my comments would be concerned with some aspects that were popular in the eighties within the field of communication engineering. With the introduction of complex modulation schemes, these aspects fled out of focus long ago, residing only in the memory of very old researchers (i.e. the ones that are not yet completely demented). These aspects are about the reliability of swapping to a memoryless model – obviously, the transition probabilities, m ones for each one of m! states, would yield quite an unreliable estimation. Then, considering the problem of estimating the event probability (refer to Techniques for Estimating the Bit Error Rate in the Simulation of Digital Communication Systems), 10006 data samples are sufficient if m = 4 and m! = 24 states/events, but not quite if m = 7 with a theoretical possibility that m! = 5040 probabilities need to be estimated.
Minor technical details:
1. Unnumbered equation above Eq. (2) comprises undefined parameter M. The option of introducing 0log0 = 0, enables counting the non-existing states, should M be a cardinal number of {S}?
2. Also, should the second summand (-H^m(x)) be squared [27]?
3. Since some well-known facts are explained in detail, for the sake of consistency, the quantity defined by the abovementioned unnumbered equation should be briefly elaborated.
4. It is great to have the figures self-explained within the captions, but the statement “the Harris limit (see main text)” is inconsistent with other explanations. Either state only “the Harris limit”, it is sufficient, or add a few words that describe it.
5. A statement from line 239 requires at least some justification – why 100, why 2?
6. In the reference [53] the journal name is missing.
7. It is more for the editorial office, but I recollect that MDPI Entropy does not support footnotes. It might have changed, but if not, the comment from the footnote should be embedded within the text (i.e. it should not be simply deleted).
Reviewer 3 Report
The manuscript explores the interesting and partially unexplored problem of how to assess variability of estimations of permutation entropy for single time series. For this, the manuscript relies on creating surrogate data replicates of the signal under analysis using the IAAFT method. The manuscript tests this approach on two synthetic signals and one real-world dataset. In the latter, the manuscript analyses brain activity to test the hypothesis that the level of entropy in eyes open decreases with time.
The evaluation is relatively limited by the number of datasets inspected but convincing in that IAAFT seems to provide reasonable estimates of the variability. The manuscript justifies the use of IAAFT on its availability and good properties to keep the power spectrum density and amplitude distribution of the signal. However, IAAFT (and surrogate data broadly) are introduced to test null hypothesis about the presence of a given property in the signal (notably, some form of nonlinear behaviour). As such, IAAFT was not originally designed to test the variability of a metric. The manuscript justifies the selection of IAAFT on its availability and good properties but I think that a more robust justification is needed in light of the fact that IAAFT is originally intended for a different use. The manuscript must compare the performance of IAAFT with that of other surrogate data methods. The manuscript could consider if simpler (e.g., random shuffling of data points) or more elaborate (e.g., iterative digitally filtered shuffled surrogates or wavelet WIAAFT surrogates) are better option. This would help to confirm that, actually, IAAFT is the right tool for the job (i.e., other surrogates - whether simpler or more complicated - do not achieve notably better performance). In particular, iterative digitally filtered shuffled surrogates and wavelet WIAAFT surrogates (both covered in the Review by Lancaster et al., 2018 cited in the paper) may be appropriate for this kind of job. IDFS surrogates account for the fact that there may be variations in the power spectrum of the signals. If that is the case, IAAFT might result in a narrow spread of values. Hence, one could envisage that IDFS may result in an estimated level of variability that is closer to that of the ensemble of signals so it could be better than IAAFT. On the other hand, WIAAFT rely on the wavelet transform. As such, it may be better suited for non-stationary signals.
On a related note, if possible, the paper must compare the surrogate-based approach with the previous art based on bootstrapping (ref [29] in the manuscript) even if only for some short time series. This would clarify which approach is more accurate. It would also be useful to compare the computational time.
Is there an effect of signal length in the accuracy of the variability estimation? The results present dependencies on parameters of the synthetic datasets but the length seems fixed.
As a minor comment, I do not think the paragraph around lines 125-131 is needed as it focuses entirely on point processes.
Round 2
Reviewer 3 Report
The authors' response has addressed and clarified the points I raised in the previous revision. I appreciate that the auhors may prefer to introduce the idea here and leave that investigation to future work. I just have a couple of minor editorial suggestions:
Lines 39 and 263 show the symbol ÷. Is this the correct one? Shouldn't it be ~ ?
I think the manuscript would benefit from a final paragraph indicating succintly the concluding message of the work. Right now, the last paragraphs focus on future work and a clear concluding paragraph is lacking.
Round 3
Reviewer 3 Report
No further comments. Congratulations on the nice piece of work.
Author Response
Thank you to the Reviewer 3 for her/his words.